# Using a Brain-Inspired Decision-Making System to Model a Real-Time Responsive Risk Assessment of the Dynamic Tasks Involved with Hazardous Materials

**Alireza Asgari * and Yvan Beauregard**

Département de Génie Mécanique, École de Technologie Supérieure, Montréal, QC H3C 1K3, Canada; yvan.beauregard@etsmtl.ca
* Correspondence: alireza.asgari.1@ens.etsmtl.ca

**Abstract:** Risk assessment of the operations utilized in processing products and services always deals with uncertainties and complexities. The ever-evolving complex and dynamic circumstances make it very difficult to identify and analyze potential events affecting workers' safety and health. Our first study was on managing the risky situations of a dynamic environment, the transport and storage of residual hazardous materials with high variation in operational times. It showed that the dynamicity of operational functions has a direct relation to the risk of accidents and suggested that such environments require a system to decide whether to perform each new action on a suspected risk condition or not. A practical framework, engaged close to the variable functions involved in potential events, is needed to provide reliable measures for risk assessment. Based on these measures, this framework would help to make decisions at the right time and to take preventive actions. It would support the decision-making process by recognizing the risk-associated features of available information and offer continuously updated alternatives for appropriate actions to prevent unsafe operations. In our second study, we developed a brain-inspired decision-making system for the real-time configuration of dynamic environments. That decision-making system builds knowledge from the least to the most similarities between experienced states to determine the most appropriate action(s) to rapidly reorient risky operations to a safe condition. This paper aims to verify the second study's proposed system performance in the simulated environment discussed in our first study on residual hazardous materials transportation. We extract information, including the effective factors, from that first study and use it in the decision-making system to prevent risky transportation. This model would be useful in daily risk management as a practical framework for establishing safe operations in today's industrial environments that involve dangerous chemical or radioactive products.

**Keywords:** residual hazardous materials; risk assessment; preventive safety actions; brain-inspired decision-making; dynamic environments



## 1. Introduction

The diversification of products and services in the marketplace has generated an unpredictable competitive environment for industries, which has forced service providers and product producers to evolve so that their processes can adjust to the dynamicity of the created environment. Dynamic environments are uncertain and their aspects are varied, not only in their intrinsic parameters but also in how those aspects are affected by externally unpredictable factors [1]. It is worth noting that the uncertainty of an environment can present both risk and opportunity for developing systems and drive the performance of the embedded processes to provide products and services at higher rates and diversities [2]. However, two of the repercussions of dynamicity and increasing operational speed are the complication of risk assessment and the shortened timeframe for taking responsive action. Risk assessment is designed to assess the risk involved in workplaces and identify the

causes of any actions that negatively affect human health and safety, e.g., actions that result in the leakage of dangerous materials, including explosive, flammable, toxic, radioactive, corrosive, or leachable materials [3]. There are some established methods, each with its own recommended tools for managing risk at the industrial level. The main risk management methods are provided by the COSO Enterprise Risk Management Framework (ERM) [2] and ISO 31000 [4].

According to the COSO ERM, risk management could maximize value if it sets objectives based on an optimum point between progress toward goals and any opposing risks. In this way, the interaction between people and organizations animates events, which could emerge as positive or negative activities with their possible opportunities and risks for the whole system. COSO ERM considers a three-dimensional matrix for the interaction between enterprise parameters and risk. It represents the relation between the goal achievement approach of the system, its structure, and relative risk management configuration [2]. ISO 31000 proposes three main approaches. One gives insight into the human-organization interrelation, and one provides a framework to implement and execute designs and to then monitor the corresponding improvements. The third approach provides maps of risk assessment and proper treatments [4].

These standards offer qualitative tools and methods, which help businesses to determine how to reach their goals and targets. Along with these approaches and standards, businesses need quantitative tools to evaluate their programs and corresponding progress toward established goals. Risks should be assessed by an appropriate framework with reliable tools and measures; still a lacuna in the approaches provided by the current standards of risk management [5,6]. The interaction between humans and organizations must be quantitively considered; the effect of human error on risk needs to be better understood, measured, and assessed [7]. The functional resonance analysis method (FRAM) was introduced as a useful tool to identify the parameters underlying operational risk through the interrelation between the functions involved in a process [8]. However, to evaluate quantitative parameters influenced by the interaction of functions and between possible effects on risk, the FRAM requires associative tools, such as the Monte Carlo simulation. We have addressed this association with the risk frameworks derived in our first study, which investigated the tasks related to the discharge, transportation, and storage of residual hazardous materials (RHMs). RHMs are substances that cause harm to people, property, and the environment [9]. Our framework, by combining the FRAM and the Monte Carlo simulation, has revealed that, alongside the availability of equipment and safety standards instruction, the dynamic aspects of the executed functions in the studied case's operation have a significant impact on meeting safety standard requirements due to their effect on increasing the risk of accidents [10].

Despite the introduction of the above-mentioned tools and methods, dynamic environments bring high uncertainty because of imperfect available information. When the data required to understand events is lacking, there is an inevitable increase in accidents. In addition, imperfect information and the associated limited understanding prevents a system from finding appropriate alternatives, leading to faulty decision-making [2]. The ability to recognize the pivotal features of information and their interactions is thus key to the success of a daily risk management framework in identifying possible operational risks [11]. With the ability to recognize risk and make the right decisions, such a system could develop its risk management performance in a dynamic environment.

We, therefore, launched our subsequent study after investigating a risk management case to develop a decision-making system that could perform with imperfect information and still make appropriate decisions. The brain has been identified as the best alternative for decision-making in environments with imperfect information, leading to the field of brain-inspired artificial intelligence (AI). The brain is nature's best adaptable system to recognize the features of imperfect information received from a dynamic environment and continuously make appropriate decisions thereabout [12] through a process of sequential evaluation and editing [13]. It generalizes knowledge based on learning from

experienced states by finding (and classifying) similarities between the features of corresponding information [12]. We, thus, introduced a brain-inspired model to process unpredictable environments and take actions with available imperfect information in our second study [14]. It mimics brain functions in decision-making by utilizing three levels of recognizing information patterns from vaguely to explicitly similar features, identifying relevant values, evaluating the known alternatives, and correspondingly making decisions. The system continuously reorganizes the available alternatives according to the new values received for the made decisions [14]. The system, after making a decision, receives that decision's added value, assigns it to the state that the decision was made for, and revaluates the existing alternatives for information in common with this state.

This paper (the third in this series) aims to verify the performance of our decision-making system on real data from our case study presented in the first paper. We simulated the case's risk assessment process to produce data for the decision-making system to identify possible risky conditions during the discharge, transportation, and store RHMs in the stocking area and then apply decisions on the process as preventive actions. It is a model for achieving a reliable level of dynamic, daily risk management. This study is a prerequisite step for such a decision-making system before can be applied for risk management in real environments.

We argue that an intelligent decision-making system will help risk management to evaluate imperfect information received from the environment in order to prevent certain events that could result in an accident. It recognizes, on a daily basis, the effective features of detected information and relative variation from a given unpredictable environment, and assigns them values that it can later utilize in making decisions to start or avoid new actions according to [14]. The continuous improvement of alternatives will produce a more trustworthy operational environment. which in turn will allow for higher levels of operational safety.

The development of our intelligent decision-making model for an adaptable, responsive risk management system for the transportation and storage of hazardous materials is presented in Section 2. We describe the experiments utilized to verify our model in daily risk management in Section 3, and then present the results of those experiments in Section 4. We analyze our results in Section 5 and present our conclusions and suggestions for future work in Section 6.

## 2. A Decision-Making System Model for Dynamic Risk Assessment

We developed a model with which to evaluate a brain-inspired decision-making system for risk management in dynamic environments. The performance of the brain-inspired decision-making system presented in our previous study [14] is tested for managing the risk involved in the case assessed by our first study [10]. We developed the model in two distinct operating environments (Figure 1): a risk assessment framework, based on a Monte Carlo simulation of the realized FRAM model of the chemical products production process in the case studied [10]; and a decision-making system, based on our work on brain-inspired decision-making [14]. In the first environment, the discharging, transportation, and storage of RHMs are simulated in two distinct parts. The first part (part-I) provides information for decision-making, which is performed in the second environment, according to the potential causes of variations in the discharge-related operations. The information gathered in part-I is thus utilized to decide if the second part (part-II) will proceed or not, based on if the transporting and storage of RHMs are determined to be at risk. The decision-making system determines if the functions in part-I (i.e., requesting a barrel for discharging RHMs, preparing the equipment, and discharging RHMs) offer a safe situation for performing part-II of each simulation state (i.e., transport and storage of RHMs). The information gathered from the simulated part-I was considered as the state of the decision-making system (the second environment). The feedback received from part-II becomes the added value of the action decided upon by the decision-making system.

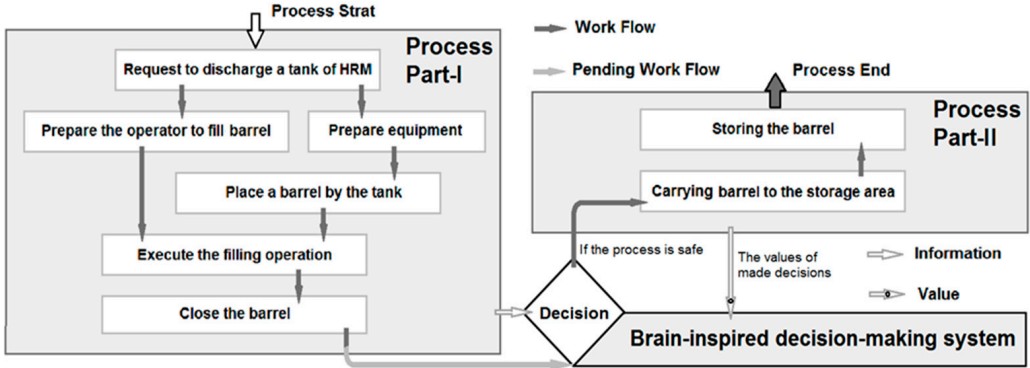

**Figure 1. Diagram of our intelligent operational risk management system in a chemical product production process.** Figure 1 shows a schematic of the complete states of the simulation. The process begins in the simulation part-I. The workflow from part-I can only proceed if the decision-making system allows the process to continue to part-II. The decision-making system receives the information it needs to decide to allow the process to continue (or not) from part-I. The added value of each decision after making reaches the decision-making system through part-II, and the system updates its alternatives accordingly.

Information is sent to the decision-making system from the simulation part-I environment via the information sensory gate, while the added values from part-II are sent to the decision-making system through the value-added sensory gate (Figure 1). The features of the information needed by the decision-making system is derived from the functions' speed-up ratios and equipment availability. The speed-up ratio defines the squeezed proportion of the time of each function according to the previous functions' delays [10]. The decision-making system processes the received information at three recognition levels. Next, a decision is made to either "Stop" or "Let the process continue to part-II". If the decision is to continue the process, the simulation's part-II continues the process and provides the feedback of corresponding added value for that decision. Accordingly, the decision-making system then modifies the relative alternatives for similar information patterns to apply to upcoming states.

### 2.1. Risk Assessment Environment

FRAM was used in [10] to identify the most important factors of functions that affect risk in a chemical production process, and then used that information to model risk assessment based on functions' speed-up ratios and equipment availability. Afterward, the model was animated and analyzed in a Monte Carlo simulation. The speed-up ratio of a function in a process is a factor that reduces the relevant scheduled time according to the functional delays applied by precedent functions. Ref. [10] considered seven parameters as direct accident influencers: unclosed-RHM-barrel, using a dolly instead of a forklift, occupied transportation path, rough floor, lack of storage space, and no available pallet. The speeds of the functions affect and intensify those direct risk influencers.

The simulated process provides the variables representing functional delays, speed-up ratios, and direct accident influencers' effects at each state. The number of accidents that occurred during a complete assessment of 25,000 states, according to [10], is the other important variable. Each simulated workday includes a chain of simulated states, and each state is the chronology of collecting, transporting, and storing RHMs barrels, in which the corresponding execution times are interrelated.

We follow the Occupational Health and Safety Regulations (OHS) three parameters [15]: performance deviation due to human error, the environmental effects on performance deviation, and the preventive actions taken against unexpected consequences, to model the process by FRAM, which has three steps:

(1)　definition of the factors of each function:

(a) input (i);
(b) output (o);
(c) resources (r) required to fulfill a function;
(d) time (t) required to complete a function;
(e) controls (c) (procedures, methods, etc.) to constrain and control each function; and
(f) preconditions (p) required to operate a function;

(2) determination of the potential variability of each function (here, the important variable is the function duration); and
(3) establishment of a network of dependencies and coupling between functions.

FRAM evaluated the potential links between function duration and failures and accidents in a chemical product production process with 11 dynamic functions and the operational chronology of functions in [10]. The following functions then were assessed:

(1) Request a barrel to discharge the RHM tank (a temporary container for conserving RHMs during the process).
(2) Prepare the transportation equipment and place the RHM barrel in front of the tank.
(3) Order the operator to wear PPE and be at in place for discharging the tank.
(4) Execute the discharging and filling of the RHM barrel.
(5) Request a label for the filled barrel.
(6) Carefully close the filled RHM barrel.
(7) Deliver and install the label on the closed barrel.
(8) Transport the closed barrel.
(9) Correctly store the closed barrel in the RHM storage area.
(10) Register the barrel.
(11) Update the registration.

Among the 11 functions listed above, the 7 that can increase delays and lead to risky conditions were selected for modeling the study's decision-making system. Those functions are modeled in two parts, as listed below for each simulation state:

- Part-I:

    1. Request a barrel to discharge the RHM tank (a temporary container for conserving RHMs).
    2. Prepare the transportation equipment and place an RHM barrel in front of the tank.
    3. Order the operator to wear PPE and be in place for discharging the tank.
    4. Execute the discharging of the RHM tank and the filling of the RHM barrel.
    5. Carefully close the filled RHM barrel.

- Part-II:

    6. Transport the closed barrel to its assigned storage area.
    7. Correctly store the closed barrel in the designated RHMs storage area.

The five functions presented in the first part may delay the process and consequently increase the speed-up ratios of the functions in the second part.

### 2.2. Decision-Making Environment

Our brain-inspired decision-making system was designed to identify the effects of the delays incurred during the first part's functions on increasing the risk of accidents during the second part's functions (transportation and storage of RHMs). The decision-making system is comprised of four sections: the information sensory gate, pattern labels' realization, alternative realization, and value-added realization (see Figure 2) to recognize and classify information through the detection of patterns and value similarities. This system uses these four sections to determine the best and worst decision alternatives decision between "Continue the process" or "Stop the process, it is a risky condition" at three levels of realization (vague, approximate, and explicit similarities). The value-added signal then receives the system's value-added sensory gate from the simulated transportation and storage of RHM (the simulation's part-II) to determine the value of the decision and records it with the detected information pattern. Next, a new alternative

replaces the best (or worst) linked alternative if the new value has a lower (or higher) score than one of them.

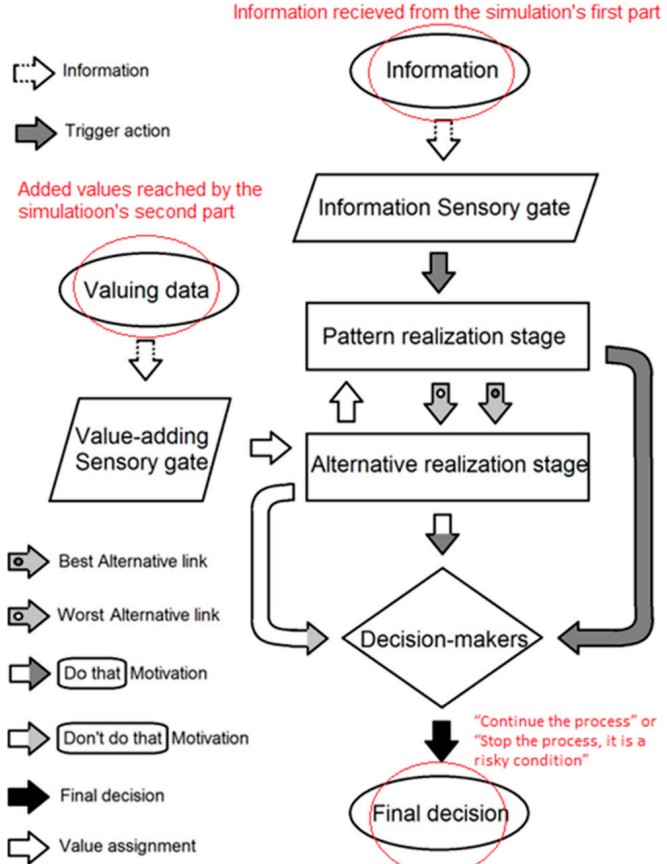

**Figure 2. Diagram of the brain-inspired decision-making system** [14]. The information-sensory gate receives a new signal from the environment, a pattern of information features (here the situation presented in the simulated part-I), and sends it to the pattern realization stage, which labels its pattern. If a label is linked to any experiences, it activates its attached alternatives at the alternative realization stage. Those alternatives determine the motivation values for the decision-making process. The decision-makers then decide to either "Continue the process" or "Stop the process, it is a risky condition". After making the decision, the value-adding sensory gate receives the added value signal from part-II and records it for the information pattern's label. If this new value suggests a new option on top of one of the best or worst alternatives, the option replaces that best (or worst) alternative on that label.

Based on the defined features of a detected state in [14], information, transmitted as a signal with eight specific frequencies, is received at the information sensory gate of the decision-making system. The signal frequencies are represented by f1, f2, f3, . . . , f8 showing the weights of each information feature. In this study, speeding-up the ratios in the first part's functions generates five of the eight information features required. "Unclosed RHM barrel" and "RHM manual transportation" are two of the remaining three features. The first five features are assigned intensities of between one and eight (the least to the highest risks) and applied in frequencies between f4 to f8. "Unclosed RHM barrel" and "RHM manual transportation" are assigned values of one (or two) sitting on f2 and f3. f1 is not assigned a value because there is no accountable feature to link to it in our information.

At each simulation state, the received signal is realized by the decision-making system before it begins the simulation's part-II so that it can decide if the process is safe to continue or not. The decision-making system decides to continue (or stop) the process based on

its knowledge. A default decision is made when there is no alternative recorded or if the difference between the best and worst alternatives is not recognizable.

The value-added achievements are extracted from the simulation's part-II to determine successful decisions. The speed-up ratios and the problematic conditions of the transportation and storage paths of part-II determine a value between 1 to 100 for a decision; this value is calculated after each "continue the process" decision to evaluate the result of that decision.

### 3. Experiment to Test the System's Performance

We built an experiment to test the decision-making system we presented in [14] using real data of risk management of the case studied in [10], considering value-added achievements (the higher the value added the higher the risk of accident) to decrease the number of accidents. This case was evaluated because it had had an especially undesirable time variation for the functions involved. Those functions are detailed in Table 1, where the "Time" feature is defined as the span required to complete a function. This case was restricted by the speed-up ratios (the ratios calculated via the previous functions' delay load). The "Control" feature shows the scheduled time for the functions.

**Table 1.** The important features of the discharge, transport, and storage of HRM functions via the FRAM in [10].

| Functions | Input | Output | Resource | Time (min) | Control | Precondition |
|---|---|---|---|---|---|---|
| request to discharge a tank of HRM | The tank is full | An empty barrel for RHM is needed | | 10 to 120 min, relative to the production process | 45 min scheduled for each request | - |
| Prepare the transportation equipment. Place the RHMs barrel in front of the tank. | An empty barrel for RHMs is needed | A dolly or forklift is ready | Prepare the transportation equipment place RHM barrel in front of the tank | An empty barrel for RHM is needed | A dolly or forklift is ready | Prepare the transportation equipment place RHM barrel in front of the tank |
| Prepare the operator to fill the barrel | An empty barrel for RHM is needed | The operator wears PPE | PPE (99.9% available) | 1 to 5 min | 3 min | Prepare the operator to fill the barrel |
| Execute the filling operation | A barrel is ready by the tank | Filling, weighing, and dating the barrel of RHM | Energy (available) | 20 to 30 min | 25 min | The operator is wearing PPE |
| Close the barrel | The RHMs barrel is filled, weighted, and dated. | A closed barrel | Air compressor (99.9% available) Energy (100%) Barrel wrench and cover (99.9% available) | 1 to 5 min | 4 min | |
| Carrying the full barrel to the storage area | A closed, labelled barrel | A closed, labeled barrel is at the storage area | Path condition "Free path and Smooth floor" (95% of the time) Energy (available) | 3 to 5 min | 4 min | The operator is wearing PPE A dolly or forklift is ready |
| Storing the barrel | A closed, labeled barrel at the storage area | A closed, labeled barrel is stored on a pallet | Storage area condition "Pallet (95% available) Space (95%)"; Energy (available) | 3 to 5 min | 4 min | The operator is wearing PPE. A dolly or forklift is ready |

For each state, the simulation began with running the simulation's part-I. The necessary information was extracted and applied as an input signal to the information sensory gate. The information features' values determine the signal shape, e.g., 12147566 (numbers that indicate the intensities of corresponding frequencies). For this pattern, the three levels of the pattern realization receive a vaguely common pattern "1211222", the approximate common pattern "12124333", and the identical one "12147566". According to the patterns received from the first part, the system would decide to continue (or stop) the simulation's part-II. The process continued in the second part if the decision was "Continue the process".

The value-added achievements are derived from the multiplication of the rank and the speed-up ratios of two functions in the simulation's part-II ("Carrying the closed barrel to the storage area" with rank one and "Storing the barrel" with rank two). The maximum value-added risk (the highest value for an accident occurrence) was equal to 100, and the minimum was one, which represents the safest condition. We trained the decision-making system with the first set of the simulation. Each simulation set included 25,000 states based on the case study. Each workday was considered as having an average of 10 consecutive states. In the second set, we evaluated the system's functioning and its effect on changing the number of accidents in the simulation.

We measured the number of successfully made decisions and compared it with the number of "Continue the process" made decisions during the experiment to verify the performance of the decision-making system. The results of four sets of simulations were collected in order to realize the system's average performance. The average recorded accidents for each set in the experiment were collected for two scenarios. In the first scenario, we assessed the system with the decision-making system, and in the second we assessed it without the decision-making system. We compared both scenarios in terms of the number of occurred accidents, with and without the decision-making system, to evaluate the utilization of the decision-making system on increasing process safety and decreasing risk.

## 4. Results

The extracted summary results from both simulation scenarios (with and without a decision-making) are gathered in Table 2. The decisions to "Continue the process" were investigated as if they were non-default decisions and they were found to produce less than 50% of the risk's added-values for similar patterns to the detected signal pattern. These were recognized as successful decisions.

**Table 2.** The simulation summary results for the first and the second parts of the simulation.

| Simulation (Average Values) Part-I | Scenario-I without Decision-Making | Scenario- II with Decision-Making |
|---|---|---|
| *Request to discharge a tank of HRMs* | | |
| The net delay of the demanding function | 24.49845701 min | 19.863123 min |
| Speed-up ratio applied to the next functions | 0.62812662 | 0.578642 |
| *Prepare the transportation equipment place RHMs barrel in front of the tank* | | |
| Is an empty barrel available? | 99.90% of the time | 99.85% of the time |
| Is a forklift available? | 64.07% of the time | 63.67% of the time |
| The net delay of the equipment preparation | 0.562222398 min | 0.706327 min |
| The equipment preparation speed-up ratio is applied to the next function | 0.64733092 | 0.53197 |
| *Prepare the operator to fill the barrel* | | |
| The net time delay to prepare the operator to fill the barrel | | |
| The operator preparation function | 0.22175735 min | 0.259966 min |
| speed-up ratio is applied to the next function | 0.63742969 | 0.535793 |
| Has the operator put on their PPE? | 62.76% of the time | 62.80% of the time |
| The added time of the previous state transporter and the transportation equipment availability | 0.00636690 min | 0.008230 min |

**Table 2.** *Cont.*

| Simulation (Average Values) Part-I | Scenario-I without Decision-Making | Scenario- II with Decision-Making |
|---|---|---|
| *Close and label the barrel* | | |
| Is the label available? | 95.01% of the time | 94.95% of the time |
| Label availability net delay | 0.14866112 min | 0.169467 min |
| The labeling function speed-up ratio | 0.73017838 | 0.633455 |
| Is the air compressor available? | 99.01% of the time | 99.04% of the time |
| Is the barrel wrench available? | 99.12% of the time | 99.08% of the time |
| Is a barrel cover available? | 98.98% of the time | 98.91% of the time |
| The net time delay of closing the barrel | 0.11720761 min | 0.134814 min |
| The barrel closing function speed-up ratio | 0.74630319 | 0.67056 |
| Is the barrel closed? | 97.11% of the time | 97.03% of the time |
| **Simulation (Average Values) Part-II** | **Scenario-I without Decision-Making** | **Scenario- II with Decision-Making** |
| *Move the closed, full barrel to the storage area* | | |
| Is the path free? | 95.02% of the time | 94.32% of the time |
| Is the floor smooth? | 95.13% of the time | 95.14% of the time |
| The net time delay for the barrel to be at the storage area | 0.53382254 min | 0.592117 min |
| The transportation speed-up ratio is applyed to the next function | 0.752859 | 0.649982 |
| Is the barrel confronted with a problematic path? | 1.29% of the time | 1.28% of the time |
| *Store the barrel* | | |
| Is a pallet available? | 94.81% of the time | 94.50% of the time |
| Is enough space available? | 95.09% of the time | 95.05% of the time |
| The net time delay for the barrel stored on a pallet | 0.30112398 min | 0.344463 min |
| The speed-up risk ratio for storing the barrel on a pallet | 0.76381236 | 0.613885 |
| **The number of accidents the barrel met in 50,000 states** | 5.5/assessment | 3.25/assessment |

The number of accidents was derived for the risk assessment of two scenarios (with and without the decision-making system). We compared both scenarios to determine the effect of the decision-making system on decreasing the probability of accidents occurring. In Table 2, scenario-I (simulation without the decision-making system) shows 5.5 accidents on average for one set of the simulation, while scenario-II (with the decision-making system) produces only 3.25 accidents in the complete session of the risk assessment. This response indicates an accident reduction of almost 40.91% is achieved with the decision-making system.

## 5. Discussion

Our study investigated an intelligent decision-making system for managing risk in a chemical product production process. We developed a model by executing a brain-inspired decision-making system presented in our previous study [14] to evaluate the risk in a simulated process based on the information gathered in [10]. The goal of this risk assessment is to prevent dangerous events and accidents from occurring. We used this model to evaluate a series of consecutive experiments in a simulated environment that explored the detailed functions involved in the discharge, transportation, and storage of RHMs. The simulated process sent information and value-added signals to the decision-making system that allowed the system to determine if the process was safe to continue (or not). At each state, the system detected an information pattern, investigated the risk level of the first part of the simulation activities in terms of their impact on the second part, and decided to order the process to "Continue the process" if it was safe, or to "Stop, the process" if the process was determined to be too high risk. Each time the decision of "Continue the process" was made, a value, calculated from the ranks and values of the second part of the simulation, was applied to the value-adding sensory gate. We then

measured the system performance through the results derived from the simulation (of each state). The results shows that the system reduced the number of accidents by 40.91%, which was very significant for the case study, as it often had a high variation in the fulfillment of scheduled tasks and a low level of equipment preparation.

The summary of the analysis of the experimental results shows that the brain-inspired decision-making system can be helpful in risk management and can contribute to the establishment of safer operational environments in dangerous environments involving chemical and radioactive hazardous materials. It also suggests that still broader insights into the utilization of brain-inspired decision-making systems may be possible for operation management in the future, such as industry 4.0 and the automation of industrial product production and service providing processes.

It should be noted that the number of variables for building information investigated here was very limited. To provide more reliable alternatives for decision-making the system needs a minimum of eight realizable features to accumulate the required information.

## 6. Conclusions

We executed a brain-inspired decision-making system to learn the hidden relationship between functions and to initiate (or halt) actions in the discharge, transportation, and storage of hazardous materials. Our model is composed of two environments, one for simulation of the process and the other for making decisions on suspected unsafe actions. Two scenarios, with and without decision-making, evaluated the decision-making system in a simulated environment. They provide a means to measure the effectiveness of the system in enhancing dynamic tasks. The experiment's results showed that a significant reduction in the number of accidents could be achieved. This study suggests that our model could be a new approach that acts as a preventive tool on risk assessment. Our model offers a different insight into using brain-inspired artificial intelligence in dynamic industrial environments and opens the way towards including a higher level of implementing artificial intelligence in today's industry. This study suggests a new method for better control of the quality and safety of operations. A future study can evaluate this investigated model in a real environment.

**Author Contributions:** Conceptualization, A.A. and Y.B.; methodology, A.A.; software, A.A.; validation, Y.B.; formal analysis, A.A.; investigation, A.A.; resources, Y.B.; data curation, A.A.; writing—original draft preparation, A.A.; writing—review and editing, A.A.; visualization, A.A.; supervision, Y.B.; project administration, Y.B.; funding acquisition, Y.B. All authors have read and agreed to the published version of the manuscript.

**Funding:** This research was funded by CRSNG, grant number RGPIN 2015-06253.

**Institutional Review Board Statement:** Not applicable.

**Informed Consent Statement:** Not applicable.

**Conflicts of Interest:** The authors declare no conflict of interest.

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
