# Peer review of "Using a Brain-Inspired Decision-Making System to Model a Real-Time Responsive Risk Assessment of the Dynamic Tasks Involved with Hazardous Materials"

_safety, 2021_

Round 1

Reviewer 1 Report

This study is dedicated to rather interesting concept based on decision making model system  similar to that  of "brain thinking" for the  real-time assessment of the risks related to  hazardous materials.  As the authors have  already emphasized, the  tool could be used   in the industrial  environments, including  chemistry  factories, engineering fields etc.

In the background description the  FRAM  tool is described and it is    concluded that  Monte Carlo  simulation is one of the key  tools, which could be used to reach the planned efficiency.

In general  the paper is well written and the idea is   well explained.

It is commendable that the introductory part provides a very precise and clear statement of the idea and substantiated benefits of the proposed model, which are clearly described in the diagram in Figure 1, showing an example in the handling and storage processes of a chemical product.

In the abstract: please describe the sentence mentioned  in page 1, lines 11-14:  "A practical framework is needed to provide a system with reliable measures to assess risk and do preventive actions. Because the dynamicity of operational functions shows having a direct relation to the risk of  accidents .."

What are the possibilities to use such method for  nuclear waste or chemical waste treatment  sectors?

How about    complicated industrial processes of  synthesis, manufacturing?

Can the developed method be  combined with the HAZOP and other  quality and safety management tools?

Author Response

Hello,

Thank you for your kind comments.  It was encouraging. I appreciate you.

About the abstract, lines 11-14 on page 1, I did some changes presented in the revised paper.  I responded to your questions in the following and I added a few sentences to the end of the section "discussion" of the paper.

What are the possibilities to use such a method for nuclear waste or chemical waste treatment sectors? How about complicated industrial processes of synthesis, manufacturing?

Yes, the goal of our studies is to increase the awareness of processes without increasing human intervention. Indeed, it can be developed for environments with high safety restrictions as nuclear waste or chemical waste treatments. Also, this model can be used as the building block by developing the information and alternative realization stages of the system for making decisions for a higher number of choices and complicated environments.

Can the developed method be combined with the HAZOP and other quality and safety management tools? It is a good idea. But, HAZDOP is a qualitative approach similar to the FRAM method, which both are worth for diagnosing a process and finding the root cause of a problem, while our system aims to act in real-time and provide an available alternative for the state that it is in to.

Sincerely

Alireza Asgari

Reviewer 2 Report

Dear authors, 

I thank you for giving me the possibility to review the paper named "Using a brain-inspired decision-making system to model a real-time responsive risk
 assessment of the dynamic tasks involved with hazardous materials". 

I find the topic very interesting even if some problems has to be underlined :

a) for literature and future research I suggest to improve your study with these papers :  https://doi.org/10.3390/app10030903 (about safety management ) and   https://doi.org/10.15866/irece.v12i2.20023 (about human factor/error). I choose the same authors for all suggestions so that you can simply find them. They use simulation through Systm Dynamics to evaluate the time/dependent trend. It is interesting .

b) about Industry 4.0 of the same authors, look at : www.doi.org/10.7232/iems.2020.19.3.551 . The author study human and Industry 4.0 in a interaction way. 

c) I think the literature is quite weak in general. Besides the paper I suggested, I think you have to improve it. 

d) The editing is poor (it's like two or more pieces taken and merged).

e) The references is not written well. Please correct them.

Author Response

Hello,

Thank you for your encouraging comments. I hope my answers would satisfy your comments and suggestions.

a) I utilized one of your indicated articles for developing the background of our study presented in the revised version.

b)  We aimed to develop the utilization of intelligent decision-making to improve industrial processes, which could be on the same page with the new age of industry i4.0. 

c) I did some changes, but it should be noted that this paper is about the implementation and test of our decision-making system (realized in our second paper) in an environment with real data, which was about risk management (presented in our first paper)

d)  We will send the article for edit as soon as possible.

e) The references are arranged.

Thank you again for your helpful comments.

Sincerely

Alireza Asgari

Reviewer 3 Report

See attached file. 

Author Response

Hello,

Thank you for your comments. I hope my answers can satisfy your facilitative comments.

The abstract hasn’t clearly stated how the method has been developed:

  • We revise it in the new version;

keywords are not that much relevant to the abstract:

  • In the revision, they are changed;

The author should explain “dynamic environments” carefully, in this paper:

  • We have explained it in the first paragraph of the introduction of the new version.

Define the meaning of “imperfect information:

  • It is explained in the second line of the fourth paragraph of the new version;

The purpose of the work should be addressed:

  • It is addressed in the fifth paragraph of the new version;

The literature is not sufficient. Intelligent modeling, big data and etc. have been widely used in dynamic modeling especially has application in the area of fault detection or risk diagnosis which is associated with the work:

  • This paper is about testing a brain-inspired decision-making system, which its advantage to the other similar models in industrial environments is defined by our second study. About the problem with the current models for risk assessment and managing risk, we explained it in the first study, which is addressed in this paper too. Here, we aimed to approach the problem with a different solution.

They haven’t addressed why the model development is needed, so it further confuses the audience to understand why two operating environments are needed to be considered:

  • We discussed it broadly in the new version (section two)

What is the diamond shape means in Figure 1? 
Figure 2 is too much afterward from the first citation.

  • The diamond shape is explained in the new figure-1. Figure-2 is adjusted in the revision;

Line 157, which seven have been selected?

  • It is revised presenting in the seventh paragraph of  section two;

Line 179 to 185 very complicated to understand. Risk is determined by the likelihood of incident arising and also the severity of consequence, see ISO31000. Is the risk determined by subjective assessment? Also line 239:

  • They are clarified in the new version;

Table 2 what is Scenario I and II means?

  • Those are revised;

Limitations of the work and implications haven’t been discussed;

  • They are explained in the new version.

Sincerely,

Alireza Asgari

Reviewer 4 Report

Dear authors, 
first of all, I would like to thank you for the opportunity to read your text.

Two main comments. Firstly it is the editing of the article that does not meet the requirements of the journal, secondly I would suggest improving the literature review. 

I would also highlight the difference between your previous publications and this one, with particular reference to the new elements. This causes some problems in defining whether an article is innovative or a minor modification of previous articles and models. 
Last but not least, I would like to emphasise that an important result is the reduction of the number of withdrawals if we use the decision-making system.

Best wishes,

Author Response

Hello, 

Thank you for your kind comments. I hope our revised paper would satisfy the addressed issues. Also, I should say this paper is the successor of our previous studies and without considering them it is not complete.
Unfortunately, some problems have existed on my side. I couldn't receive the review results. This week I get access to them. I did the necessary revisions. I developed the literature. I will send the paper for thorough editing too.

Sincerely

Alireza Asgar

Round 2

Reviewer 2 Report

Congratulations

Author Response

Hello,

Thank you. I appreciate your time.

Reviewer 3 Report

The authors have addressed my concerns. Just remind of use MDPI-Safety's template. 

Author Response

Hello,

Now, I've revised the paper and put it in MDPI-Safety's template.

Thank you. I appreciate your time.

Reviewer 4 Report

Dear Authors,

thank you very much for all additional explanation. It was great pleasure to read your work.

Best wishes,

Author Response

The paper is edited for English and put into MDPI-Safety's template.

Thank you. I appreciate your time.